# Ultrasound-Compatible Electrode for Functional Electrical Stimulation

**DOI:** 10.3390/biomedicines12081741

**Published:** 2024-08-02

**Authors:** Sunho Moon, Xiangming Xue, Vidisha Ganesh, Darpan Shukla, Benjamin C. Kreager, Qianqian Cai, Huaiyu Wu, Yong Zhu, Nitin Sharma, Xiaoning Jiang

**Affiliations:** 1The Department of Mechanical and Aerospace Engineering, North Carolina State University, Raleigh, NC 27606, USA; smoon4@ncsu.edu (S.M.); dshukla2@ncsu.edu (D.S.); bkreage@ncsu.edu (B.C.K.); qcai2@ncsu.edu (Q.C.); hwu15@ncsu.edu (H.W.); yzhu7@ncsu.edu (Y.Z.); 2The Joint Department of Biomedical Engineering, North Carolina State University, Raleigh, NC 27695, USA; xxue5@ncsu.edu (X.X.); vganesh3@ncsu.edu (V.G.); nsharm23@ncsu.edu (N.S.); 3The Joint Department of Biomedical Engineering, University of North Carolina at Chapel Hill, Chapel Hill, NC 27599, USA

**Keywords:** body-conforming ultrasound-compatible electrode, functional electrical stimulation (FES), silver nanowire (AgNW) electrode, ultrasound (US), wearable ultrasound

## Abstract

Functional electrical stimulation (FES) is a vital method in neurorehabilitation used to reanimate paralyzed muscles, enhance the size and strength of atrophied muscles, and reduce spasticity. FES often leads to increased muscle fatigue, necessitating careful monitoring of the patient’s response. Ultrasound (US) imaging has been utilized to provide valuable insights into FES-induced fatigue by assessing changes in muscle thickness, stiffness, and strain. Current commercial FES electrodes lack sufficient US transparency, hindering the observation of muscle activity beneath the skin where the electrodes are placed. US-compatible electrodes are essential for accurate imaging and optimal FES performance, especially given the spatial constraints of conventional US probes and the need to monitor muscle areas directly beneath the electrodes. This study introduces specially designed body-conforming US-compatible FES (US-FES) electrodes constructed with a silver nanowire/polydimethylsiloxane (AgNW/PDMS) composite. We compared the performance of our body-conforming US-FES electrode with a commercial hydrogel electrode. The findings revealed that our US-FES electrode exhibited comparable conductivity and performance to the commercial one. Furthermore, US compatibility was investigated through phantom and in vivo tests, showing significant compatibility even during FES, unlike the commercial electrode. The results indicated that US-FES electrodes hold significant promise for the real-time monitoring of muscle activity during FES in clinical rehabilitative applications.

## 1. Introduction

Movement disorders comprise a range of neurological conditions that can lead to abnormal movements, limb weakness, or complete paralysis. These disorders may result from conditions such as spinal cord injury (SCI), stroke, or Parkinson’s disease (PD). SCI and stroke are impactful neurological injuries that can result in paralysis. The Traumatic Spinal Cord Injury Facts National Spinal Cord Injury Statistical Center reports that the United States experiences approximately 18,000 new cases of SCI diagnosed annually, with an estimated 302,000 people living with SCI in the country [1,2]. Stroke was identified by the American Stroke Association as the fifth leading cause of death and disability in the United States, with a substantially decreased quality of life [3]. Furthermore, PD is a progressive neurodegenerative disorder characterized by a notable loss of dopaminergic neurons in the substantia nigra, leading to motor dysfunctions such as tremor and gait difficulties [4,5]. Tremors, involuntary muscle oscillations in the hands and arms, affect over 11 million individuals in the United States [6,7]. Functional electrical stimulation (FES) is frequently prescribed as a therapeutic technique to ameliorate the symptoms of movement disorders by preserving or increasing muscle size to prevent atrophy, as well as facilitating the functional movement of a paretic or even paralyzed muscle [3,8] or suppressing the pathological movements associated with conditions such as PD [9]. FES can be applied to muscles through surface electrodes that are detached from the skin after use. Additionally, FES can promote neuroplasticity and motor relearning by facilitating repetitive movement patterns and enhancing motor recovery outcomes [10,11]. Despite these benefits, the rapid onset of FES-induced muscle fatigue limits the efficacy of FES, resulting in swift loss of FES-elicited muscle force.

Multiple monitoring modalities are currently employed, including electromyography (EMG), ultrasound (US) imaging, magnetic resonance imaging (MRI), etc. Among them, surface electromyography (sEMG) is commonly used as a non-invasive technique for monitoring muscle activity. However, sEMG is restricted by factors such as a low signal-to-noise ratio (SNR), the capacity to solely monitor superficial muscle activity, the degradation of signal due to fatigue, and the influence of sensor positioning on sensitivity [12,13]. Furthermore, the existence of stimulation artifacts during FES poses a challenge to simultaneously measure EMG signals and perform muscle stimulation [14]. Within the medical field, US imaging stands out as a widely utilized non-invasive technique for investigating the morphological structure of tissue. It has found extensive applications in assessing various aspects including muscle size, thickness, fatigue, and treatment effectiveness [15,16,17]. In addition, US serves as a reliable indicator of muscle quality, encompassing parameters such as strength, size, volume, and physical function [18,19].

The use of US imaging offers numerous advantages, effectively overcoming the inherent limitations of surface electromyography (sEMG). Initially, it provides high-resolution visualization of deep tissues, enabling real-time monitoring of muscle activities during FES without data distortion caused by stimulation artifacts or muscle crosstalk [15,20]. Furthermore, real-time US imaging objectively assesses and visualizes soft tissue morphologies, eliminating muscle crosstalk during contractions and enhancing the effectiveness of biofeedback in optimizing rehabilitation exercises [21]. Additionally, US imaging with flexible sparse array transducers allows for dynamic visualization of muscle activities, including the relative displacement of muscle groups during body movements [22]. Lastly, in contrast to sEMG, which measures force, US imaging provides insight into the spatial characteristics of muscle movement [23,24,25]. Moving from the elucidation of the benefits of US imaging to technological advancements in rehabilitation, the integration of real-time US imaging with one-degree-of-freedom (1-DOF) hybrid knee exoskeletons represents a significant progression in rehabilitation technology [26]. This integration features a closed-loop control system consisting of an FES system and a powered orthosis [26]. By reliably identifying muscle fatigue, it optimizes rehabilitation protocols and minimizes the likelihood of overexertion risks [26]. At present, the commercially available FES electrode cannot be used for the simultaneous monitoring of targeted muscle activity induced by FES and US imaging due to the lack of US transparency. Understanding the importance of monitoring muscle states in FES applications, the development of FES electrodes compatible with US imaging is a pressing imperative. These specialized electrodes allow US transducers to be placed directly on top of them, enabling precise muscle monitoring in confined spaces. Given the necessity to monitor muscle states and the evident advantages of US imaging, a compelling motivation exists to develop US-compatible FES (US-FES) electrodes for accurate muscle monitoring during targeted FES implementation.

Building upon the preliminary study, we investigated the feasibility of using the US-FES electrode composed of a silver nanowire/polydimethylsiloxane (AgNW/PDMS) composite as a flexible conductive electrode [27,28]. The layer of AgNW functions as the FES electrode for muscle stimulation, with the addition of PDMS serving to mitigate electromagnetic interference from FES without causing US imaging distortion. The sheet resistance and electrical conductivity of the US-FES electrode were first characterized, and its performance was validated by comparing wrist bending angles induced by FES with both proposed and commercial hydrogel FES electrodes under various electrical stimulation parameters such as current amplitude, frequency, and pulse width. Next, A-mode US signal detection was utilized to assess the feasibility of US-FES electrode evaluation for monitoring muscle fibers. The spatial resolution of B-mode US imaging in both lateral and axial dimensions was calculated to compare differences with and without the US-FES electrode. Finally, a wearable US array was integrated with US-FES electrodes for real-time imaging during rehabilitation in vivo. The results show that our US-FES electrodes are capable of effectively capturing A-line US signals and B-mode images, a capability not found in commercial electrodes.

## 2. Materials and Methods

### 2.1. Ultrasound-Compatible FES (US-FES) Electrode Design, Fabrication, and Characterization

Silver nanowires (AgNWs) were synthesized by a modified polyol process [29]. Briefly, 60 mL of 0.147 M polyvinylpyrrolidone (PVP) (Mv∼40,000; Sigma-Aldrich, St. Louis, MO, USA) in ethylene glycol (EG) was added to a round-bottomed flask suspended in an oil bath that was heated to 151.5 °C and magnetically stirred at 260 rpm. After 1 h, 200 μL of a 24 mM CuCl_2_ (CuCl_2_·2H_2_O, 99.999+%; Alfa Aesar, Haverhill, MA, USA) solution in EG was added and heated for 15 min. Lastly, 60 mL of 0.094 M AgNO_3_ (99+%; Sigma-Aldrich, St. Louis, MO, USA) solution in EG was added, and the reaction was left to proceed for 2 h. The products were washed first with acetone and then several times with ethanol. The produced nanowires had an average length and diameter of approximately 25 μm and 100 nm, respectively. AgNWs in ethanol were drop-casted on a pre-cleaned glass slide with a patterned mask with a diameter of 3.2 cm made by laser cutting Kapton tape. The ethanol was evaporated at 50 °C and the process was repeated several times, producing a thickness of 3.56 ± 0.39 μm for the AgNW film. The sheet resistance and electrical conductivity of the AgNW film were 0.154 ± 0.018 Ω sq^−1^ and (1.840 ± 0.220) × 10^6^ S m^−1^, respectively. Next, liquid PDMS (Sylgard 184, Dow Inc., Midland, MI, USA) in the weight ratio of 10:1 of prepolymer to cross-linker was degassed and spin-coated at 400 rpm for 30 s on the AgNW film. After curing at 50 °C for 3 h, the AgNW/PDMS electrode was peeled off from the glass slide, revealing surface-embedded AgNWs in PDMS. A cable from a commercial FES electrode (PALS^®^ Electrodes, Axelgaard, Fallbrook, CA, USA) was attached to the AgNW/PDMS electrode using silver paste to connect to the current-controlled stimulator (Rehastim1, HASOMED GmbH, Magdeburg, Germany). The fabrication process of the AgNW/PDMS electrode is schematically shown in Figure 1. The thickness of the AgNW film was measured using a Keyence VKX1100 confocal laser scanning microscope. A 4-probe method was used to determine the resistance of the AgNW film. The sheet resistance (R_s_) was calculated using the formula R_s_ = (Rw)/l, where R, w, and l represent the resistance, line width, and length of the AgNW film, respectively. The electrical conductivity (σ) was calculated using the formula σ = l/(RA), where l, R, and A represent the length, resistance, and cross-sectional area of the AgNW film, respectively.

### 2.2. Evaluation of the Effectiveness of US-FES Electrode during FES Applications

All experimental procedures involving human subjects in this study were approved by the Institutional Review Board (IRB) protocol 19247 at North Carolina State University. A commercial hydrogel electrode (PALS^®^ Electrodes, Axelgaard, Fallbrook, CA, USA) was selected for the comparison study. The electrode is 3.2 cm in diameter. The round electrode designed for electrical muscle stimulation (EMS) consists of layers of a patented stainless-steel mesh fabric and hydrogel. To mitigate the potential muscle fatigue from FES, a three-second stimulation duration was introduced to measure the peak response [30]. A lower stimulation duration initially achieved the maximum wrist angle for the given FES parameters. Approximately one minute was allowed between tests for muscle recovery from the FES-induced fatigue. This interval was crucial as it ensured each testing session began with a relatively consistent baseline, minimizing cumulative fatigue effects on experimental outcomes.

The experimental setup for the validation test is depicted schematically in Figure 2a. Tests were conducted on the right forearm. The FES electrodes were placed 5 cm from the lateral condyle on the forearm and 5 cm from the carpals in the tendinous region, as illustrated in Figure 2a. Stimulation applied to the forearm extensors induced wrist extension. An IMU (Yost Labs Inc., Portsmouth, OH, USA) was used to measure the wrist extension angle induced by FES. A state of relaxation corresponded to a wrist bend angle of 0°. The angular wrist change was assessed following FES using a current-controlled stimulator (Rehastim1, HASOMED GmbH, Magdeburg, Germany). This study aimed to assess the effects of different stimulation parameters, specifically current amplitude (5, 8, and 11 mA), stimulation frequency (20, 30, and 40 Hz), and pulse duty cycle (20%, 40%, and 60%). The 3 s FES was replicated four times.

### 2.3. Ultrasound Compatibility Testing for US-FES Electrode with Single-Element Ultrasound Transducer

The experimental setup for the feasibility examination using A-mode US detection is illustrated in Figure 2b. A 3.5 MHz single-element US transducer was employed to estimate the insertion loss of both the US-FES electrodes and commercial electrodes. The single-element US transducer was positioned vertically in direct alignment with the electrodes, and US gel (Aquasonic 100, Parker Lab Inc., Fairfield, NJ, USA) was applied. The pulser/receiver (5077 PR, Olympus, Redmond, WA, USA) was connected to the transducer and operated at a pulse repetition frequency (PRF) of 200 Hz and an input voltage of 200 V with a bandpass filter ranging from 1 to 10 MHz. The oscilloscope (DSO7104B, Agilent Technologies, Santa Clara, CA, USA) was used to display the radiofrequency (RF) echo signal reflecting from the muscle fibers.

For wrist extensor stimulation, FES electrodes were placed on the upper part of the forearm and the tendinous region, then connected to a current-controlled stimulator (Rehastim1, HASOMED GmbH, Magdeburg, Germany). The stimulation parameters included current amplitudes at 5, 8, and 11 mA, frequencies at 20, 30, and 40 Hz, and pulse duty cycles at 20%, 40%, and 60%. The B-mode US imaging of the target muscle was initially collected using a wireless ultrasound probe (L7, Clarius, Vancouver, BC, Canada) to determine a reference location for the A-mode echo signal from the single-element US transducer. The time of flight (TOF) of A-mode signals was calculated and compared to the B-mode images. Subsequently, echo signals were obtained both with and without the US-FES electrode to determine the extent of insertion loss. Lastly, the single-element transducer was positioned above both the customized and commercial electrodes to evaluate the feasibility of collecting echo signals while the FES was taking place.

### 2.4. Ultrasound Compatibility Test of US-FES Electrode with Ultrasound Array

To establish a reference for the A-mode US signals captured with a single-element US transducer, initially, B-mode imaging of the target muscle was obtained to predict the muscle’s reflective layers during wrist extensor relaxation and contraction. A schematic representation of the experimental setup for the feasibility test using B-mode US images is illustrated in Figure 3. The examination employed a wireless ultrasound probe (L7, Clarius, Vancouver, Canada) operating at 7 MHz. This setup was devised to evaluate the US compatibility of both the customized and commercial electrodes. A lab-fabricated linear US array transducer using 32 elements and measuring 10 mm × 5.5 mm with a center frequency of 7 MHz was specifically designed to explore the adaptability of the US-FES electrode for integration with wearable ultrasonic devices.

The linear array was vertically positioned directly on top of the electrodes to showcase the US compatibility of the US-FES electrode, akin to the feasibility test involving A-mode US. The Verasonics ultrasound system (Vantage 256, Kirkland, WA, USA) was connected for signal transmission and reception to the array. The RF data were acquired and used to reconstruct real-time B-mode images of the target muscle with the software MATLAB 2021b (MathWorks, Natick, MA, USA). The commercial US probe (L7, Clarius, Vancouver, BC, Canada) also acquired real-time B-mode US images of the target muscle, providing additional assessment of the US compatibility of both electrode types. To demonstrate the reliability of merging US-FES electrodes with wearable devices, real-time B-mode images were collected using both the in-house linear array and commercially available US probe to monitor the movement of the specific muscle of interest during FES.

### 2.5. Phantom Preparation for Resolution Comparison Depending on the Presence of US-FES Electrode during B-Mode Imaging

A wire phantom was created for measuring the axial and lateral resolutions of the transducer. Firstly, a mold was prepared using copper wire measuring a diameter of 160 μm. The wires were affixed within the phantom mold. The wires ranged in axial depth from 8 mm to 11 mm, with approximately 1.5 mm spacing between them. The lateral separation between this set of wires was approximately 3.5 mm. Axial depth pertains to the distance between the transducer aperture and the wire, while axial separation is the distance between adjacent wires in the direction parallel to the transducer aperture. After bonding the wires in the mold, gelatin was prepared using a hot plate, a magnetic stir rod, degassed water, and type A, 300-Bloom gelatin derived from acid-cured porcine skin (G2500, Sigma-Aldrich Corp., St. Louis, MO, USA). Degassed water was warmed to 49 °C on the hotplate. While stirring rapidly, 5% m/V of gelatin was added to the water. Once the solution became optically transparent, we allowed it to cool for several minutes and transferred the mixture to the mold. The mold was refrigerated for 24 h during the cross-linking process of the mixture.

### 2.6. Feasibility Assessment of US-FES Electrodes for Tremor Patients

A single participant diagnosed with Parkinson’s disease (male, 67 years old) participated in this study. The aim was to assess the effectiveness of tremor suppression by using US-FES electrodes in conjunction with real-time B-mode US imaging during the electrical stimulation process. The participant was seated and instructed to perform two trials of a grasping motion, involving the action of picking up and holding a cup in response to a visual cue after 5 s. The commercial hydrogel FES electrodes (PALS^®^ Electrodes, Axelgaard, Fallbrook, CA, USA) and US-FES electrodes were strategically placed on the participant’s Flexor carpi radialis (FCR) forearm. The application of stimulation commenced 15 s after initiation, with parameters of a 150 Hz frequency, 4 mA amplitude, and a 200 μs pulse width. The linear US transducer (Prodigy, S-Sharp, New Taipei City, Taiwan) was positioned above the electrodes, and an IMU (Yost Labs Inc., Portsmouth, OH, USA) was placed at the wrist joint. IMU signals were sampled at a frequency of 1000 Hz and synchronized with US plane wave images. A center frequency of 5 MHz and a sampling frequency of 20 MHz were used for US images, captured at 1000 fps using an external trigger programmed in MATLAB/Simulink, and implemented on a real-time target machine (Speedgoat, Liebefeld, Bern, Switzerland). During data collection, IMU signals were continuously recorded in real time, while the RF data of the US images were saved for each trial. The experimental setup is illustrated in Figure 4.

## 3. Results

### 3.1. Effectiveness of US-FES Electrode

To evaluate the effectiveness of the US-FES electrode (the optical image and structure of the US-FES electrode are shown in Figure 5), a comparative test was conducted against a commercial FES electrode. Figure 6 shows the impact of electrical stimulation from the US-FES electrode on the wrist extensor compared to the performance of the commercial FES electrode. Different stimulation parameters were modified, including the current amplitude, frequency, and pulse duty cycle of electrical stimulation within the ranges of 5 mA to 11 mA, from 20 Hz to 40 Hz, and from 20% to 60%, respectively. Notably, a current amplitude of 5 mA was found to be the sub-threshold for inducing a response on both electrodes. Regarding performance, the US-FES electrode exhibited a comparably minor variance compared to the commercial hydrogel electrode. The data on peak wrist bend measurements induced by FES are compiled in Table 1, based on four separate tests. We also present representative raw time-series data (angular acceleration, angular velocity, and angular angle) collected by the IMU during muscle extension under different stimulation parameters (20 Hz, 8 mA, 40%; 20 Hz, 11 mA, 40%; 40 Hz, 8 mA, 40%; 20 Hz, 8 mA, 60%), as shown in Figure 7. Moreover, we summarize other characteristics of the time-series data, such as the time delay between the beginning of extension and stimulation onset and the rate of angle change, in Table 2. Additionally, we visually compared the wrist angles stimulated by the PALS^®^ and US-FES electrodes based on the data in Table 1 (as shown in Figure 8a). To support the effectiveness of FES using our US-FES electrodes, we conducted two-sample t-tests for each set of stimulation parameters by comparing the wrist contraction angles. As depicted in Figure 8a, the *p*-values were greater than 0.05 for most parameter sets, suggesting no significant difference in wrist angles induced by PALS^®^ and US-FES electrodes. Notably, two parameter sets of stimulation (30 Hz, 8 mA, 40%; 20 Hz, 8 mA, 60%) showed statistically significant differences, with *p*-values of 0.0039 and 0.0005, respectively, indicating that the US-FES can be significantly more effective than the commercial electrode. Moreover, we compared the time delay between the onset of FES and the starts of muscle extension (Figure 8b) as well as the rate of angle change (Figure 8c). These characteristics can influence the effectiveness of muscle fiber recruitment as outcome measures. The analysis of representative time-series angle data demonstrates that our US-FES electrode proves significant FES effectiveness comparable to that of the commercial electrode.

### 3.2. Feasibility of Ultrasound-Compatible FES Electrode with A-Mode Ultrasound

Figure 9a illustrates B-mode US images of the target muscle in states of relaxation and contraction of the wrist extensor, with the muscle layers serving as regions of interest. In the relaxed state, these muscle layers were situated approximately 1.50 cm and 2.10 cm below the skin surface, respectively. Following FES, the layers moved to 1.8 cm and 2.30 cm from the skin surface, respectively. Referring to the B-mode US images of the targeted muscle, the A-mode US signal from the muscle layers in the targeted muscle was collected and is shown in Figure 9b. The initial echo (a) from the upper muscle layer occurred at 20.60 μs, corresponding to a depth of 1.58 cm calculated from the TOF. The subsequent echo (b) at 27.25 μs is equivalent to 2.10 cm when in a relaxed state. During muscle contraction, the echo (c) at 23.75 μs indicated a depth of 1.83 cm. The following echo (d) observed at 30.30 μs indicated a distance of 2.30 cm, which closely aligned with the location of muscle layers seen in B-mode imaging.

To assess the insertion loss of the US-FES electrode (depicted in Figure 10), changes in peak-to-peak voltage (V_pp_) were measured. Without the US-FES electrode, the V_pp_ of the first and second echo signals were 1.48 V and 2.23 V, respectively. When the US-FES electrode was placed vertically in front of the transducer, the V_pp_ of the first and second echoes from the muscle layers were 0.83 V and 2.1 V. Although the first echo experienced a reduction of −5.02 dB, the following echo decreased by –0.52 dB compared to the one without the US-FES electrode. Next, the performance of the US-FES electrode was assessed against the commercial FES electrode. Using the commercial electrode could not yield a US signal from the muscle layers. However, the echo signal from the muscle layers was successfully captured with the US-FES electrode. The findings demonstrated that the specially designed US-FES electrode was compatible with ultrasound, enabling the transducer to effectively capture signals in the FES region.

### 3.3. Feasibility of US-Compatible FES Electrode with B-Mode Imaging

We initially assessed the US penetration capability of the US-FES electrode using a commercially available ultrasound probe (L7, Clarius, Canada) to obtain B-mode US images of the target muscle. Figure 11 represents the B-mode US image of the forearm muscle under three conditions: without the FES electrode (Figure 11a), with the US-FES electrode (Figure 11b), and with the PALS^®^ commercial electrode (Figure 11c). As shown in Figure 11b, the area where the US-FES electrode was placed revealed an acoustic shadow in the B-mode images, attributed to the presence of the PDMS layer of the US-FES electrode. However, this shadow does not have a substantial impact on the quality of B-mode images, as the layers of the wrist extensor muscle remain clearly visible. In contrast, when the PALS^®^ FES electrode was present, parts of the B-mode image were obstructed, therefore obscuring the visualization of target muscle anatomy. Only areas lacking the presence of the commercial electrode display a normal B-mode image. Thus, the US-FES electrode has the potential to be applied to biomedical US imaging for FES applications.

Furthermore, B-mode images were reconstructed using a lab-fabricated linear US array and Verasonics US research system, without the FES electrode, with the US-FES electrode, and with a commercial FES electrode, as shown in Figure 12. Consistent with the findings from B-mode US images obtained by the Clarius probe, the brightness of the image can be reduced through the insertion of the US-FES electrode shown in Figure 12b, though not significantly affecting B-mode images quality. However, no discernible features appeared in the B-mode US image when the commercial electrode was positioned below the US array. This result underscores the excellent potential for using the US-FES electrode in biomedical applications that require both simultaneous US imaging and FES.

To quantitatively analyze the impact of the presence of the US-FES electrode on B-mode US imaging performance, the spatial resolution of the B-mode image was compared in Figure 13. Figure 13b shows the wire phantom images using a lab-fabricated US linear array with and without a US-FES electrode situated beneath the array. The wire placed at a depth of 8.5 mm was selected for tests comparing spatial resolution. In the absence of the US-FES electrode, the lateral and axial resolutions of the wires measure 837.8 μm and 567.5 μm, respectively. In the presence of the US-FES electrode, the lateral and axial resolution are 972.9 μm and 648.6 μm, respectively. The lateral and axial resolution were sacrificed by 16.8% and 14.3%, respectively, when the US-FES electrode was present.

To verify the feasibility of the US-FES electrode, B-mode US imaging was used to monitor target muscle movement in response to FES while the US-FES electrode was placed directly beneath the linear US transducer. The US-FES electrode was compared to a commercial FES electrode. As shown in Figure 14, B-mode images of the wrist extensor muscle activity were collected during FES. With FES activated, contraction of the wrist extensor occurs as shown in the B-mode images depicting the movement of muscle layers away from the skin surface. According to Figure 14a, the US-FES electrode successfully stimulated the muscle to induce contraction, while the US array monitored the movement of the targeted muscle without significant image distortion. However, the B-mode image of muscle anatomy cannot be provided when utilizing a commercial electrode, as shown in Figure 14b. This test demonstrated that the US-FES electrode is effective for applications requiring both US imaging and FES concurrently.

### 3.4. Tremor Suppression Ratio Calculation and B-Mode Image Evaluation

To assess tremor suppression during the tests, a tremor suppression ratio ($T_{r}$) is computed using the following formula:(1)$$T_{r}=1-\frac{rms(\omega_{tr})}{rms(\omega_{b})}$$
where, *rms* represents the root mean squared value of a specific signal and $\omega_{tr}$ is the angular movement of the wrist around the vertical, as measured by the IMU during periods in which stimulation was turned on, and $\omega_{b}$ is the angular velocity during baseline periods without any stimulation. The tremor suppression ratios obtained from the two trials are listed in Table 3. A representation of collected B-mode images is depicted in Figure 15.

## 4. Discussion

The tests revealed that the AgNW/PDMS US-FES electrode is compatible with ultrasound imaging. Unlike commercial FES electrodes, the PDMS layer of the customized US-FES electrode effectively shielded the electromagnetic interference (EMI) when the US transducer was vertically oriented. We initially applied Tensive^®^ conductive adhesive gel (Parker Laboratories, Inc., Fairfield, NJ, USA) to the AgNW side of the electrode to adhere the electrode to the skin. Although the gel helped attach the electrode to the desired area, its application was untidy and challenging to remove after the experiment. It also caused irritation to the patient’s skin upon removal. Due to the disadvantages, double-sided dermal sensing hydrogel tape (Axelgaard, Fallbrook, CA, USA) was selected for its ability to adhere effectively to the skin and the US-FES electrode without causing irritation following FES testing for several hours. Subsequent works will include comparing various coupling medium candidates in terms of their impact on ultrasound imaging performance.

The single-element US transducer requires pressure to receive echo signals from the target muscle even amidst FES. The presence of pressure on the skin may potentially affect the positioning of the FES electrodes with the intended stimulated area, despite the application of the adhesive gel. Referring to Figure 9, the reductions in Vpp for the first and second echoes at 44% and 6%, respectively, could be attributed to transducer alignment with each muscle layer. Thus, this emphasizes the potential necessity for a wearable single-element ultrasound transducer and arrays that can not only eliminate the pressure requirement on the transducers for data collection, but also mitigate concerns related to misplacement of both FES electrodes and US transducers. A shadow in the B-mode US imaging caused by the US-FES electrode’s placement is present in Figure 11b. This issue can be resolved by reducing the PDMS layer thickness of the US-FES electrode to improve US penetration or adjusting its acoustic impedance by adding a chemical compound such as aluminum oxide.

Spatial resolution is an important consideration for quantitatively evaluating the imaging performance of US transducers. We calculated both axial and lateral resolution. The axial resolution primarily depends on the frequency, which remains unaffected by the imaging depth, while the lateral resolution can be influenced by multiple components such as frequency, the aperture size of the transducer, the focal depth of ultrasound, beamwidth, side lobes, and grating lobes [31]. The thickness of the US-FES electrode at 200 μm can improve the imaging depth, and the focal depth of the ultrasound transducer placed on top of the US-FES electrode can be adjusted. As a result, the spatial resolution can increase across a dynamic range of 0 to −6 dB.

Since the US-FES electrode will be developed for integration with wearable US devices in the future, a prototyped linear array transducer has been designed and fabricated for vertical orientation with the US-FES electrode. Our integrated system provides a novel approach to neurorehabilitation by combining FES with real-time US imaging of muscles. This integration allows clinicians not only to stimulate muscle activation, but also to assess the structural integrity and biomechanical performance of muscles, tendons, surrounding tissues, etc., thereby optimizing treatment strategies and enhancing functional recovery. The discourse encompasses limitations inherent in previous studies, emphasizing the imperative for more comprehensive evaluation. In this paper, we can highlight that the US-FES electrode induced a higher wrist contraction angle with the same size as the commercial FES electrode, indicating greater effectiveness of FES. Moreover, the US-FES electrode has been successfully proven to be feasible for B-mode US imaging of muscle activity in the FES-stimulated target area, demonstrating high potential for biomedical applications requiring simultaneous US imaging and FES. Furthermore, we will develop an electromyography (EMG) sensor that is combined with a flexible US transducer for more accurate and precise muscle activity monitoring. The multi-modal sensor technology possesses a significant potential for applications in neurorehabilitation and assistive robotics.

## 5. Conclusions

This study introduced a AgNW/PDMS US-FES electrode developed to be US-compatible, and the effectiveness of the electrical stimulation electrode was evaluated across various stimulation parameters. The assessment revealed a performance akin to that of the commercial hydrogel FES electrodes. Importantly, when positioned vertically beneath the US transducer (both the single-element transducer and array transducer), the US-FES electrode demonstrated acoustic matching that allowed the transducer to capture echo signals from the forearm muscle layers with minimal insertion loss. The findings strongly suggest promising prospects for the US-compatible FES electrode in precise muscle activity monitoring in the intended FES-stimulated region.

## Figures and Tables

**Figure 1 biomedicines-12-01741-f001:**
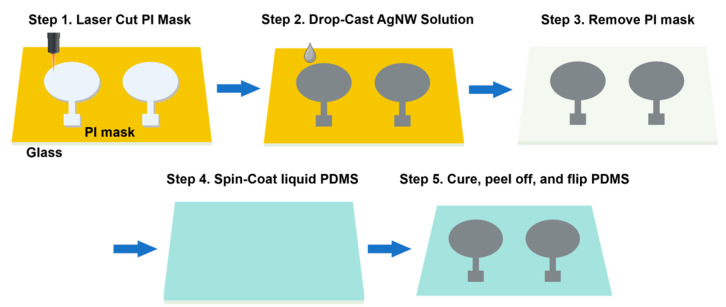
A schematic describing the AgNW/PDMS electrode fabrication process.

**Figure 2 biomedicines-12-01741-f002:**
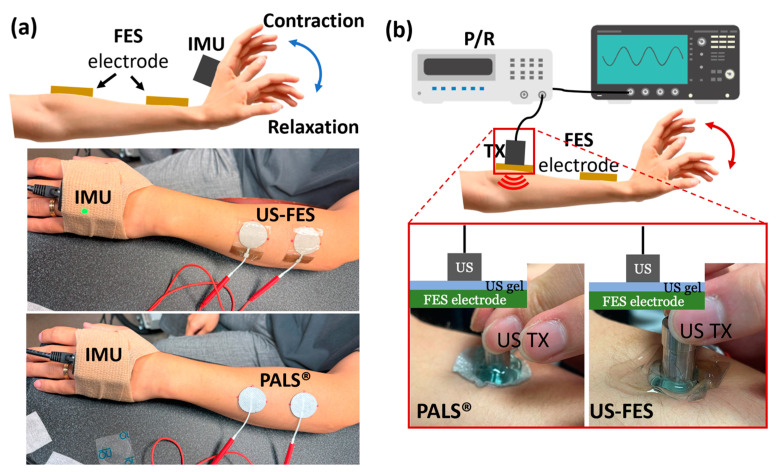
(**a**) Experimental setup for evaluation of the effectiveness of the US-FES electrode. While the wrist extensor is contracted by FES, an IMU tracks the angle of the wrist. The setup includes a commercial FES electrode (PALS^®^) and the US-FES electrode. (**b**) Experimental setup for the feasibility test of the US-FES electrode using a 3.5 MHz single-element US transducer. The US transducer collects A-mode US signals from the muscle to assess if the electrode is compatible with ultrasound. The feasibility test involves placing the US transducer on top of both the commercial FES electrode and the US-FES electrode.

**Figure 3 biomedicines-12-01741-f003:**
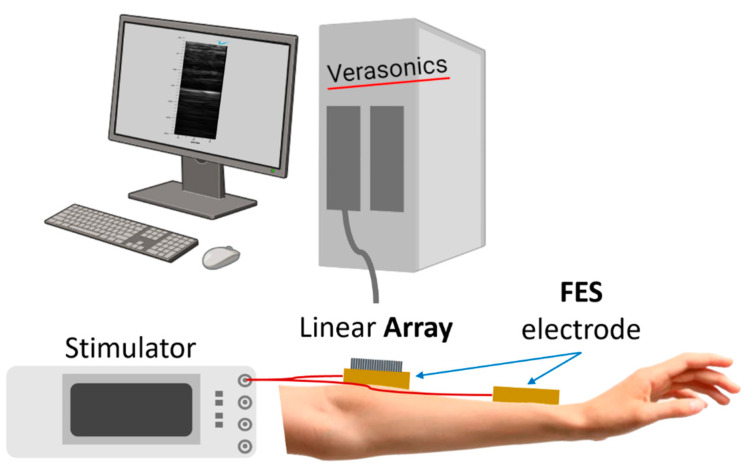
Schematic of the experimental setup for evaluating the feasibility of the US-FES electrode with linear US array transducers. A commercial linear array probe (L7, Clarius, Vancouver, BC, Canada) and a lab-fabricated linear array transducer operating around 7 MHz were used to assess the feasibility of integrating the US-FES electrode with a wearable ultrasonic device.

**Figure 4 biomedicines-12-01741-f004:**
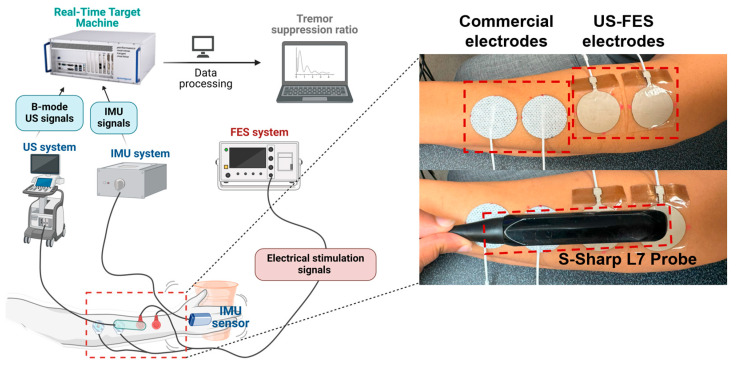
Experimental setup for data collection with patients experiencing tremors.

**Figure 5 biomedicines-12-01741-f005:**
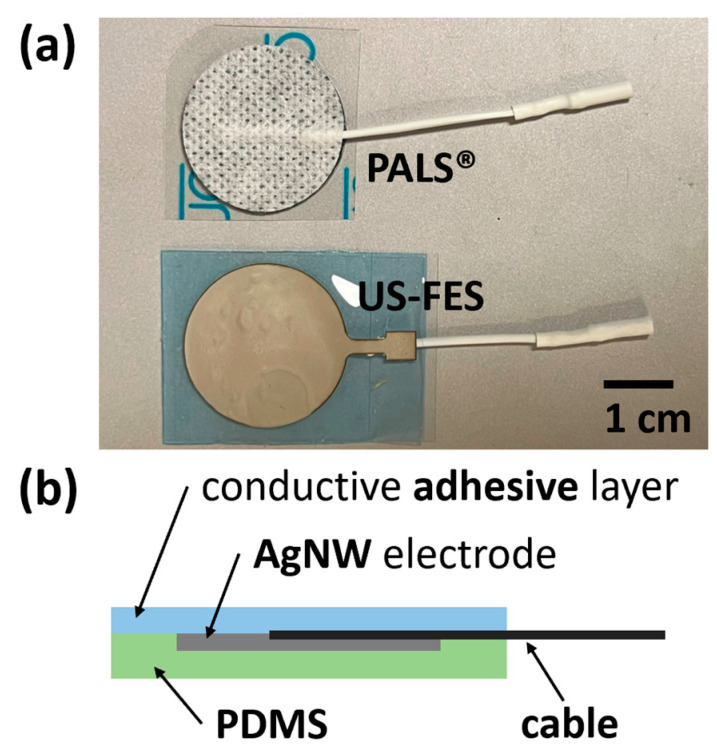
(**a**) The US-compatible FES (US-FES) electrode and commercial FES electrode (PAL^®^); (**b**) the schematic representation of the electrode’s structure.

**Figure 6 biomedicines-12-01741-f006:**
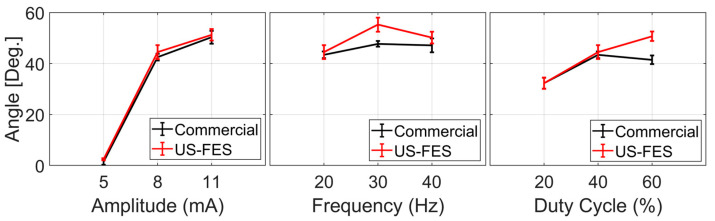
Evaluation of the effectiveness of electrical stimulation using the US-FES electrode on the wrist extensor response compared to the performance of the commercial FES electrode (PALS^®^), taking into account various stimulation parameters: current amplitude (5–11 mA), frequency (20–40 Hz), and pulse duty cycle (20–40%).

**Figure 7 biomedicines-12-01741-f007:**
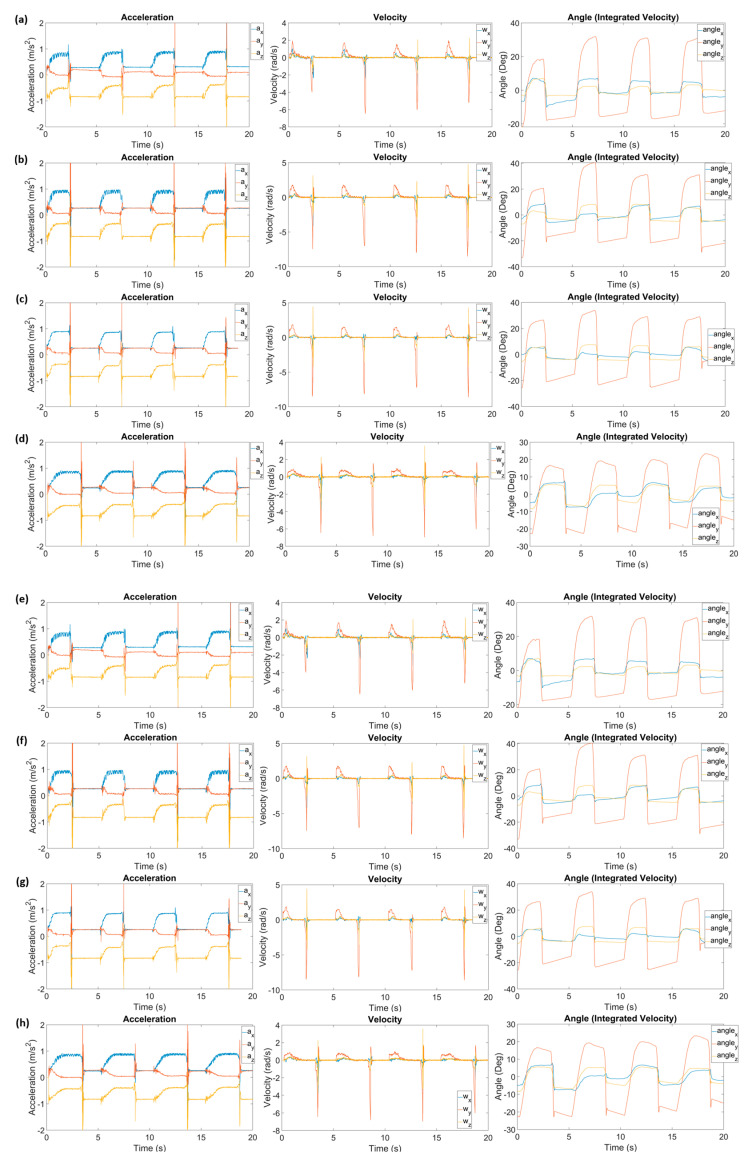
Representative time-series data (angular acceleration, angular velocity, and angular angle) collected by IMU during muscle extension induced by FES (stimulation onset at 0, 5, 10, and 15 s) using PALS^®^ (**a**–**d**) and US-FES electrode (**e**–**h**) with different stimulation parameters: (**a**,**e**) 20 Hz, 8 mA, 40% (Base); (**b**,**f**) 20 Hz, 11 mA, 40%; (**c**,**g**) 40 Hz, 8 mA, 40%; (**d**,**h**) 20 Hz, 8 mA, 60%.

**Figure 8 biomedicines-12-01741-f008:**
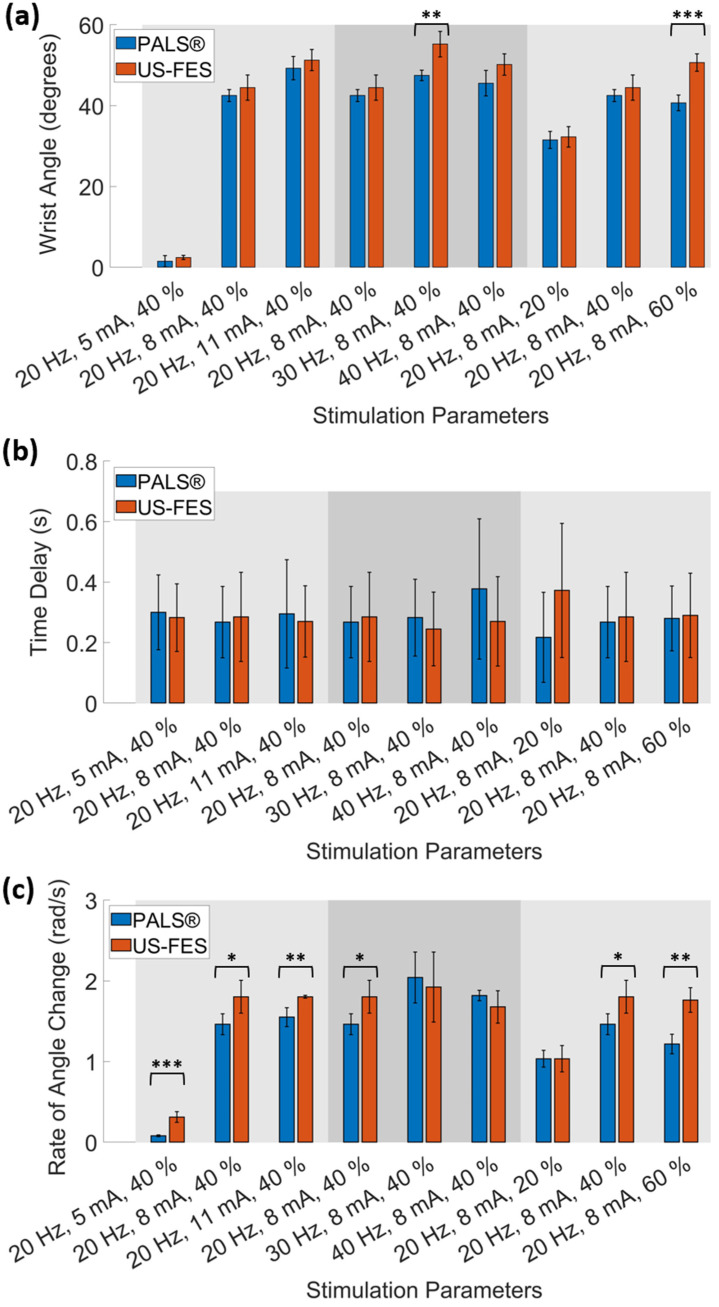
(**a**) Comparison of wrist extension angles, (**b**) the time delay between stimulation onset and the start of the muscle extension, and (**c**) rate of angle changes measured by IMU using different stimulation parameters (* for *p* < 0.05, ** for *p* < 0.01, and *** for *p* < 0.001).

**Figure 9 biomedicines-12-01741-f009:**
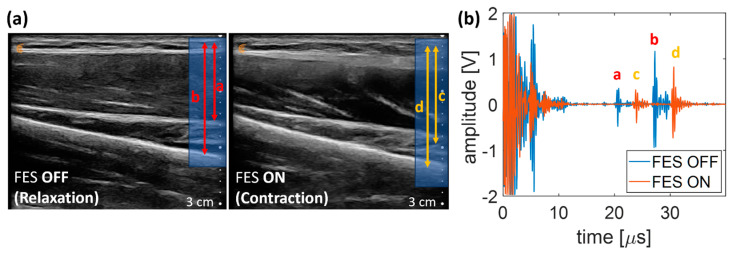
B-mode ultrasound imaging of target layers of the muscle (**a**) as a reference for the status of relaxation and contraction of the wrist extensor (a and c, b and d are same targeted muscle layers, respectively). (**b**) A-mode echo signals with the single-element ultrasound transducer to assess the matching of TOF with layers of muscle as presented in B-mode ultrasound imaging (echo a–d are corresponding to the target layers of the muscle in B-mode US images from Figure 9a).

**Figure 10 biomedicines-12-01741-f010:**
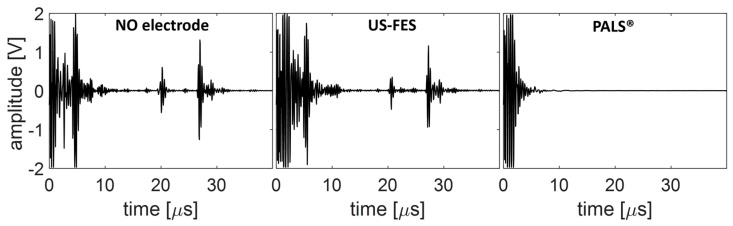
Changes in peak-to-peak voltage were measured to assess the insertion loss induced by the customized electrode (US-FES) and commercial FES electrode (PALS^®^) in front of a US transducer.

**Figure 11 biomedicines-12-01741-f011:**
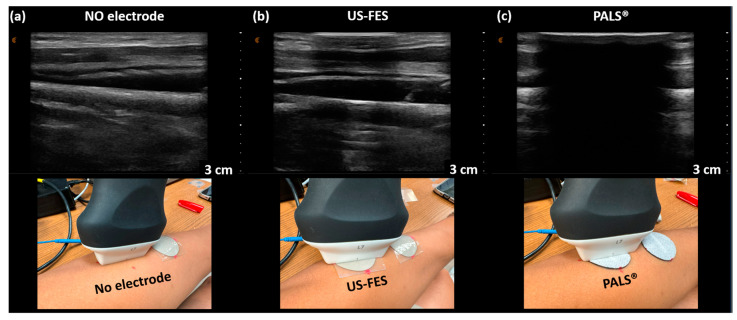
B-mode US imaging of the wrist extensor using a commercial ultrasound probe (Clarius, Vancouver, BC, Canada). US images of the layers of the muscle were (**a**) collected without any electrode present, (**b**) with the customized US-FES electrode, and (**c**) with the commercial FES electrode placed in front of the ultrasound probe to compare their effects on image quality. The position of the ultrasound probe is shown in the second row of pictures.

**Figure 12 biomedicines-12-01741-f012:**
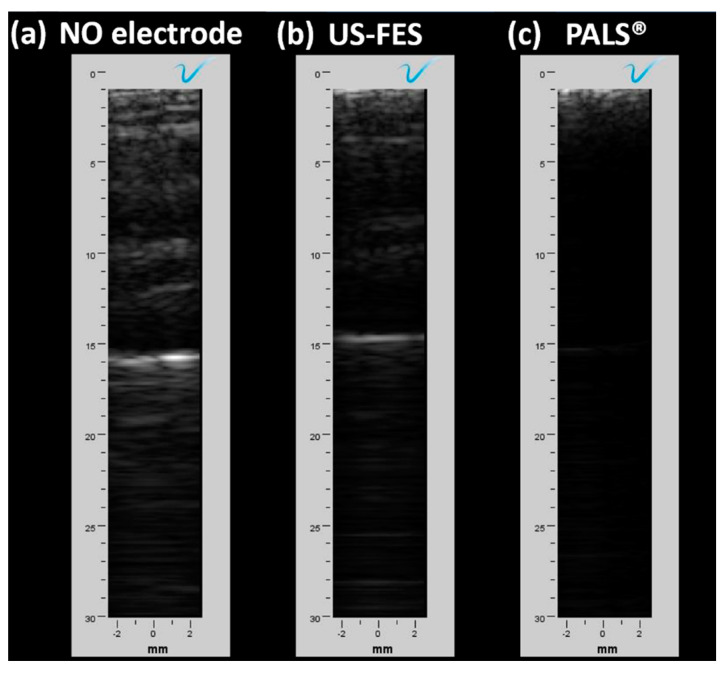
B-mode US imaging of the wrist extensor using a lab-fabricated linear US array transducer and Verasonics US system. US images of the layers of the muscle were collected (**a**) without any electrode present, (**b**) with the customized US-FES electrode, and (**c**) with the commercial FES electrode placed in front of the ultrasound probe to compare their effects on image quality.

**Figure 13 biomedicines-12-01741-f013:**
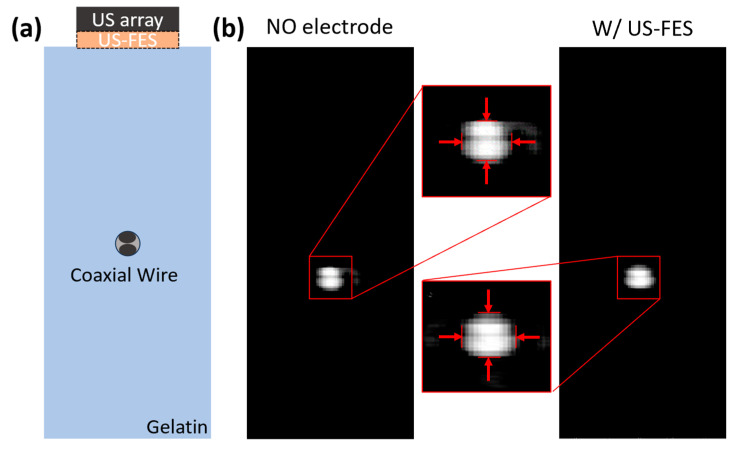
The lateral and axial resolution tests depending on the presence of the US-FES electrode. (**a**) The schematic of the phantom using coaxial cable. (**b**) The B-mode image of the phantom with the dynamic range from 0 to −6 dB (arrows from left and right represents lateral resolution and arrows from top and bottom indicated axial resolution).

**Figure 14 biomedicines-12-01741-f014:**
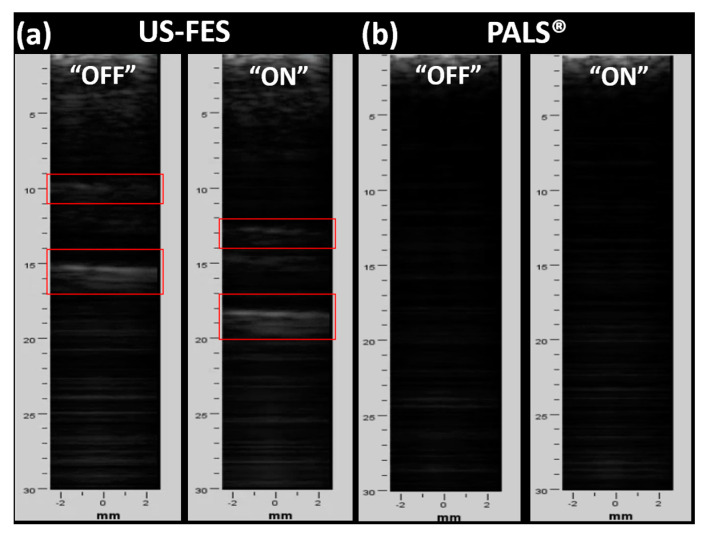
B-mode US imaging of wrist extensors using a lab-fabricated linear ultrasound array transducer and Verasonics ultrasound system during FES. US images of the layers of the muscle were collected to evaluate if B-mode images can be effectively obtained during FES using (**a**) the US-FES and (**b**) commercial FES electrode (red frame represents layers of targeted muscle).

**Figure 15 biomedicines-12-01741-f015:**
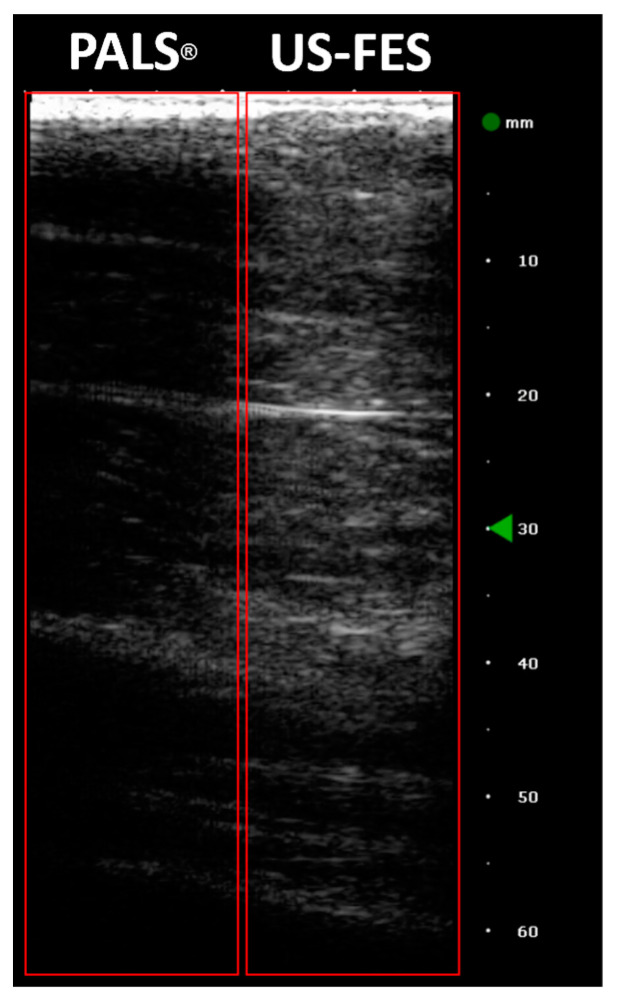
B-mode images from one trial with PALS^®^ and US-FES electrodes (Red frame on left and right are the area where PALS^®^ and US-FES electrode are placed, respectively. Green triangular arrow on the depth axis indicates the focus depth).

**Table 1 biomedicines-12-01741-t001:** Wrist angle measurements for four trials (#1, #2, #3, and #4 named in table) in comparison to the performance of the commercial FES electrode (PALS^®^) using different stimulation parameters: current amplitude (5–11 mA), frequency (20–40 Hz), and pulse duty cycle (20–40%) of electrical stimulation.

Parameters		Wrist Angle (PALS^®^)					Wrist Angle (US-FES)				
		#1	#2	#3	#4	Ave.	#1	#2	#3	#4	Ave.
Amplitude	20 Hz, 5 mA, 40%	3.58°	0.97°	0.61°	0.76°	1.48°	2.79°	2.79°	2.35°	1.74°	2.42°
	20 Hz, 8 mA, 40%	44.43°	40.79°	42.28°	42.45°	42.49°	40.12°	47.36°	44.4°	45.94°	44.46°
	20 Hz, 11 mA, 40%	52.83°	48.78°	49.63°	45.84°	50.28°	53.68°	53.34°	49.03°	48.99°	51.26°
Frequency	20 Hz, 8 mA, 40%	44.43°	40.79°	42.28°	42.45°	43.40°	40.12°	47.36°	44.4°	45.94°	44.46°
	30 Hz, 8 mA, 40%	47.87°	47.01°	46.01°	49.05°	47.70°	58.7°	55.56°	55.57°	51.04°	55.22°
	40 Hz, 8 mA, 40%	47.15°	40.82°	46.88°	47.29°	47.12°	52.61°	46.93°	49.06°	52.04°	50.16°
Pulse Duration	20 Hz, 8 mA, 20%	34.52°	31.47°	30.14°	29.92°	32.28°	32.43°	35.12°	32.63°	28.93°	32.28°
	20 Hz, 8 mA, 40%	44.43°	40.79°	42.28°	42.45°	43.40°	40.12°	47.36°	44.4°	45.94°	44.46°
	20 Hz, 8 mA, 60%	41.66°	38.59°	39.54°	42.87°	41.43°	51.69°	50.69°	47.65°	52.56°	50.65°

**Table 2 biomedicines-12-01741-t002:** The measurement of the time delay between the start of muscle extension and FES onset and the rate of angle change for each trial ((#1, #2, #3, and #4 named in table) in comparison to the performance of the commercial FES electrode (PALS^®^) using different stimulation parameters: current amplitude (5–11 mA), frequency (20–40 Hz), and pulse duty cycle (20–40%) of electrical stimulation.

**Parameters**		**Time Delay (PALS^®^)**					**Time Delay (US-FES)**				
		**#1**	**#2**	**#3**	**#4**	**Ave.**	**#1**	**#2**	**#3**	**#4**	**Ave.**
Amplitude	20 Hz, 5 mA, 40%	0.13 s	0.30 s	0.35 s	0.42 s	0.30 s	0.13 s	0.27 s	0.35 s	0.38 s	0.28 s
	20 Hz, 8 mA, 40%	0.11 s	0.26 s	0.31 s	0.39 s	0.27 s	0.11 s	0.23 s	0.35 s	0.45 s	0.29 s
	20 Hz, 11 mA, 40%	0.07 s	0.25 s	0.37 s	0.49 s	0.30 s	0.11 s	0.27 s	0.31 s	0.39 s	0.27 s
Frequency	20 Hz, 8 mA, 40%	0.11 s	0.26 s	0.31 s	0.39 s	0.27 s	0.11 s	0.23 s	0.35 s	0.45 s	0.29 s
	30 Hz, 8 mA, 40%	0.12 s	0.26 s	0.33 s	0.42 s	0.28 s	0.10 s	0.21 s	0.28 s	0.39 s	0.25 s
	40 Hz, 8 mA, 40%	0.10 s	0.29 s	0.49 s	0.63 s	0.38 s	0.11 s	0.20 s	0.32 s	0.45 s	0.27 s
Pulse Duration	20 Hz, 8 mA, 20%	0.05 s	0.16 s	0.26 s	0.40 s	0.22 s	0.13 s	0.26 s	0.47 s	0.63 s	0.37 s
	20 Hz, 8 mA, 40%	0.11 s	0.26 s	0.31 s	0.39 s	0.27 s	0.11 s	0.23 s	0.35 s	0.45 s	0.29 s
	20 Hz, 8 mA, 60%	0.14 s	0.26 s	0.33 s	0.39 s	0.28 s	0.11 s	0.25 s	0.40 s	0.40 s	0.29 s
**Parameters**		**Rate of Angle Change (PALS^®^)**					**Rate of Angle Change (US-FES)**				
		**#1**	**#2**	**#3**	**#4**	**Ave.**	**#1**	**#2**	**#3**	**#4**	**Ave.**
Amplitude	20 Hz, 5 mA, 40%	0.09 rad/s	0.07 rad/s	0.07 rad/s	0.09 rad/s	0.08 rad/s	0.37 rad/s	0.22 rad/s	0.34 rad/s	0.32 rad/s	0.31 rad/s
	20 Hz, 8 mA, 40%	1.27 rad/s	1.53 rad/s	1.53 rad/s	1.52 rad/s	1.46 rad/s	1.95 rad/s	1.71 rad/s	1.56 rad/s	1.99 rad/s	1.80 rad/s
	20 Hz, 11 mA, 40%	1.44 rad/s	1.66 rad/s	1.64 rad/s	1.46 rad/s	1.55 rad/s	1.82 rad/s	1.78 rad/s	1.80 rad/s	1.81 rad/s	1.80 rad/s
Frequency	20 Hz, 8 mA, 40%	1.27 rad/s	1.53 rad/s	1.53 rad/s	1.52 rad/s	1.46 rad/s	1.95 rad/s	1.71 rad/s	1.56 rad/s	1.99 rad/s	1.80 rad/s
	30 Hz, 8 mA, 40%	1.57 rad/s	2.19 rad/s	2.18 rad/s	2.23 rad/s	2.04 rad/s	1.68 rad/s	2.57 rad/s	1.76 rad/s	1.68 rad/s	1.92 rad/s
	40 Hz, 8 mA, 40%	1.85 rad/s	1.85 rad/s	1.85 rad/s	1.72 rad/s	1.82 rad/s	1.87 rad/s	1.53 rad/s	1.48 rad/s	1.83 rad/s	1.68 rad/s
Pulse Duration	20 Hz, 8 mA, 20%	0.91 rad/s	0.99 rad/s	1.11 rad/s	1.13 rad/s	1.04 rad/s	1.03 rad/s	1.26 rad/s	0.97 rad/s	0.88 rad/s	1.04 rad/s
	20 Hz, 8 mA, 40%	1.27 rad/s	1.53 rad/s	1.53 rad/s	1.52 rad/s	1.46 rad/s	1.95 rad/s	1.71 rad/s	1.56 rad/s	1.99 rad/s	1.80 rad/s
	20 Hz, 8 mA, 60%	1.37 rad/s	1.09 rad/s	1.16 rad/s	1.25 rad/s	1.22 rad/s	1.91 rad/s	1.55 rad/s	1.82 rad/s	1.77 rad/s	1.76 rad/s

**Table 3 biomedicines-12-01741-t003:** Tremor suppression ratio ($T_{r}$) evaluation.

Trial #	$\mathrm{rms}\;(\omega_{b}$)	$\mathrm{rms}\;(\omega_{tr}$)	$T_{r}$ (%)
**1**	13.3746	8.6476	35.34
**2**	10.6083	3.5944	66.11

## Data Availability

The data that support the findings of this study are available from the corresponding author, upon reasonable request.

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
