# Peer review of "Ultrasound-Compatible Electrode for Functional Electrical Stimulation"

_biomedicines, 2024, doi:10.3390/biomedicines12081741_

Round 1

Reviewer 1 Report

Comments and Suggestions for Authors

The purpose of this study was to investigate the feasibility of the developed ultrasound-comparable FES (US-FES) electrode. The authors evaluated the developed US-FES electrode by comparing with a commercial FES electrode. The authors demonstrated that B-mode ultrasound images were reconstructed through the US-FES electrode, which was not possible through the commercial FES electrode. In general, the topic of the current manuscript may be interesting for the readers in this field. There are several issues to be addressed:

1. The size and shape are different between the US-FES electrode and the commercial FES electrode. It is likely that the effective electrical potential field depends on the various factors including the size, shape, and location of the electrodes, possibly resulting in a different recruitment of muscle fibers. These factors might be attributable to the measurable difference between the two electrodes given that the variability of the measures is very small. It seems feasible that the authors could make the electrodes comparable in dimensions. Please clarify.

2. It seems that the authors used one IMU sensor to estimate hand segment orientation. It is important to analyze the time-series angle data because the stimulation parameters can also influence the effectiveness of the recruitment of muscle fibers. It is suggested to demonstrate the representative time-series angle data for each condition and to add more detailed characteristics of the time-series angle data (e.g., the delay between the stimulation and onset, the rate of angle change, etc.) as outcome measures.

3. There is no statistical result in the current manuscript. It is suggested that the appropriate statistical tests will further support the stimulation effectiveness of the developed US-FES electrodes compared to the commercial FES electrodes.

4. It would be good to include a schematic to describe the electrode fabrication process.

5. In general, the example B-mode ultrasound images do not show muscle architectural information clearly. Moreover, it seems that the images in Figure 8 were collected from the different locations. It would be good to image the muscle from the same location across conditions.

Author Response

Comments and Suggestions for Authors (Reviewer 1)

The purpose of this study was to investigate the feasibility of the developed ultrasound-comparable FES (US-FES) electrode. The authors evaluated the developed US-FES electrode by comparing with a commercial FES electrode. The authors demonstrated that B-mode ultrasound images were reconstructed through the US-FES electrode, which was not possible through the commercial FES electrode. In general, the topic of the current manuscript may be interesting for the readers in this field. There are several issues to be addressed:

Comments 1. The size and shape are different between the US-FES electrode and the commercial FES electrode. It is likely that the effective electrical potential field depends on the various factors including the size, shape, and location of the electrodes, possibly resulting in a different recruitment of muscle fibers. These factors might be attributable to the measurable difference between the two electrodes given that the variability of the measures is very small. It seems feasible that the authors could make the electrodes comparable in dimensions. Please clarify.

Response 1: We appreciate the reviewer’s comment regarding the different size and shape of the electrodes between US-FES and the commercial FES. It is plausible that these factors might affect the effective electrical potential field and eventually muscle fiber recruitment. In order to address the reviewer’s concern, we have reconducted the following:

  1. We have redesigned and fabricated US-FES electrodes with same dimensions as those of the commercial FES electrodes (Diameter of 3.2 cm)
  2. We conducted additional tests with newly fabricated US-FES electrode to compare the performance with commercial FES electrode

Results suggest that our US-FES electrode with the same dimension as the commercial electrode demonstrates effective electrical stimulation and comparable muscle fiber recruitment. We have revised the tables and figures with the new experimental data in the manuscript (See table 1 and Figure 6). Thank you for your feedback to have a more robust evaluation of our US-FES electrode.

Figures revised by using the newly fabricated US-FES electrode with the same dimension as commercial electrode (Figure 2 (a), 4, and 5 (a)):

Comments 2. It seems that the authors used one IMU sensor to estimate hand segment orientation. It is important to analyze the time-series angle data because the stimulation parameters can also influence the effectiveness of the recruitment of muscle fibers. It is suggested to demonstrate the representative time-series angle data for each condition and to add more detailed characteristics of the time-series angle data (e.g., the delay between the stimulation and onset, the rate of angle change, etc.) as outcome measures.

Response 2: We thank the reviewer for suggesting the importance of analyzing time-series angle data to demonstrate the effectiveness of the muscle fiber recruitment induced by stimulation. To address this concern, we have undertaken the following:

  1. Time-series angle data were collected by IMU sensor during muscle extension induced by FES.
  2. We plotted the representative time-series angle data, acceleration data, and velocity data for the conditions (20Hz, 8mA, 40% (Base); 20Hz, 11mA, 40%; 40Hz, 8mA, 40%; 20 Hz, 8 mA, 60%), included below.

Results show a more comprehensive understanding of how parameters of FES affect the recruitment of muscle fiber. The plots below show the time series angle data, acceleration data, and velocity data during the four trials of extension as outcome measures. Thank you for the reviewer’s suggestion, which has improved our analysis.

Figure 7. Specify representative time-series data (acceleration, velocity, and angle) collected by IMU during muscle extension with different stimulation parameters. (a) & (e) 20 Hz, 8 mA, 40 % (Base); (b) & (f) 20 Hz, 11 mA, 40 %; (c) & (g) 40 Hz, 8 mA, 40 %; (d) & (h) 20 Hz, 8 mA, 60%

Comments 3. There is no statistical result in the current manuscript. It is suggested that the appropriate statistical tests will further support the stimulation effectiveness of the developed US-FES electrodes compared to the commercial FES electrodes.

Response 3: The reviewer’s suggestion to include statistical results to support the stimulation effectiveness of US-FES electrodes is valuable. In response, we have conducted a series of statistical tests, including the computation of the average and standard deviation (Fig. 8 (a)), as well as two-sample t-tests (Fig. 8 (b)) for each set of stimulation parameters (amplitude, frequency, and pulse duration) to compare the wrist contraction angles induced by the two types of electrodes. The results of these tests indicate the following:

    1. For most parameter sets, the p-values were greater than 0.05, indicating no significant difference between the wrist angles induced by PALS® and US-FES electrodes.
    2. However, two parameter sets showed statistically significant differences (p-values < 0.05):
      • 30 Hz, 8 mA, 40 %: US-FES was significantly more effective than PALS® (p-value = 0.0039).
      • 20 Hz, 8 mA, 60 %: US-FES was significantly more effective than PALS® (p-value = 0.0005).

These results suggest that, while the overall effectiveness of the developed US-FES electrodes is comparable to the commercial FES electrodes, there are specific conditions where US-FES demonstrates significantly higher stimulation effectiveness. We have included these detailed statistical analyses and results in the revised manuscript, providing a more comprehensive evaluation of the US-FES electrodes' performance.

Additionally, we have included two plots (fig. 8 (a) and (b)) below that visually compare the wrist angles induced by the PALS® and US-FES electrodes across different parameter sets with average, and standard deviation and display the p-values to indicate the statistical significance of the differences observed. Background colors and labels have been added to differentiate between the amplitude, frequency, and pulse duration parameter sets.

Figure 8. (a) comparison of wrist extension angles induced by FES with PALS® and US-FES electrode using different stimulation parameters (b) The p-values for two-sample t-test between PALS® and US-FES electrode

Comments 4. It would be good to include a schematic to describe the electrode fabrication process.

Response 4: We agree with the reviewer to add a schematic describing the electrode fabrication process. We have included a detailed schematic of the electrode fabrication process in the revised manuscript (Figure 1). This schematic shows each step of the fabrication process to obtain the US-FES electrode. Thank you for your suggestion, which has enhanced the clarity of our manuscript.

Figure 1. A schematic describing the AgNW/PDMS electrode fabrication process

Comments 5. In general, the example B-mode ultrasound images do not show muscle architectural information clearly. Moreover, it seems that the images in Figure 8 were collected from the different locations. It would be good to image the muscle from the same location across conditions.

Response 5: We appreciate the reviewer’s concerns about B-mode ultrasound images we collected from the different locations. To clarify, we tried to put ultrasound probe at the same location as much as we can. To ensure consistency, we marked the skin in red at the monitoring site, allowing us to keep the ultrasound probe at the same position accurately for each measurement as you can see in the picture below. However, we understand that the ultrasound gel may have caused subtle movements during muscle extension, which could make some of variations observed in the images. Thank you for this valuable investigation.

Figure 11.

Reviewer 2 Report

Comments and Suggestions for Authors

This manuscript details the development of ultrasound-compatible functional electrical stimulation (FES) electrodes and investigates their potential for clinical application, specifically targeting the enhancement of neurorehabilitation techniques. The introduction effectively provides comprehensive background information on the limitations of current FES electrodes and the potential benefits of ultrasound-compatible electrodes, including relevant references that contextualize the study within the broader field of neurorehabilitation and biomedical engineering. The study design is appropriately aligned with the objective of comparing the performance of the newly developed AgNW/PDMS electrodes with that of a commercial hydrogel electrodes. The methodology covers the synthesis of silver nanowires, electrode fabrication, and detailed experimental setups for both in vitro and in vivo experiments, ensuring high replicability. The results are clearly presented, demonstrating that the new electrode exhibits comparable conductivity and performance to commercial electrodes, with superior ultrasound compatibility as evidenced in phantom and in vivo tests. This supports the goal of enabling real-time monitoring of muscle activity during FES. Overall, the manuscript provides a novel solution to an important problem in neurorehabilitation and is a significant contribution to the field in compliance with the standards of the journal, Biomedicines.

Author Response

This manuscript details the development of ultrasound-compatible functional electrical stimulation (FES) electrodes and investigates their potential for clinical application, specifically targeting the enhancement of neurorehabilitation techniques. The introduction effectively provides comprehensive background information on the limitations of current FES electrodes and the potential benefits of ultrasound-compatible electrodes, including relevant references that contextualize the study within the broader field of neurorehabilitation and biomedical engineering. The study design is appropriately aligned with the objective of comparing the performance of the newly developed AgNW/PDMS electrodes with that of a commercial hydrogel electrodes. The methodology covers the synthesis of silver nanowires, electrode fabrication, and detailed experimental setups for both in vitro and in vivo experiments, ensuring high replicability. The results are clearly presented, demonstrating that the new electrode exhibits comparable conductivity and performance to commercial electrodes, with superior ultrasound compatibility as evidenced in phantom and in vivo tests. This supports the goal of enabling real-time monitoring of muscle activity during FES. Overall, the manuscript provides a novel solution to an important problem in neurorehabilitation and is a significant contribution to the field in compliance with the standards of the journal, Biomedicines.

Response: 

We appreciate the reviewer’s positive feedback on our manuscript. Your comments affirm the value of our US-FES electrodes in advancing neurorehabilitation techniques.

We are particularly pleased that you acknowledge the potential clinical applications of our AgNW/PDMS electrodes with ultrasound compatibility, which aligns with our goal of enabling real-time US imaging of muscle activity during FES with our electrode.

Again, thank you for your positive feedback. Your comments have been incredibly motivating and valuable to us.

Round 2

Reviewer 1 Report

Comments and Suggestions for Authors

Thank you for the authors’ effort to address the previous comments. The revised manuscript has been improved in many aspects. There are some suggestions:

1. In the revised manuscript, the final joint angles are about 50° which is considerably smaller compared to the original manuscript. Was there any change in the experiment protocol? Please clarify.

2. In Figure 7, why did the authors use the joint angles calculated from angular velocity data? Since the authors have IMU data, the orientation of the IMU sensor can be calculated. Furthermore, it would be good to compare the orientations between the commercial FES electrodes and the developed US-FES electrodes. It is suggested to demonstrate the representative time-series angle data for each condition (i.e., the commercial vs. the developed electrodes) and to add more detailed characteristics of the time-series angle data (e.g., the delay between the stimulation and onset, the rate of angle change, etc.) as outcome measures in Table.

3. Figure 8b (i.e., the bar plot for p values) is not required. Asterisk (*) can be used to indicate p < 0.05 in Figure 8a.

Author Response

Thank you for the authors’ effort to address the previous comments. The revised manuscript has been improved in many aspects. There are some suggestions:

  1. In the revised manuscript, the final joint angles are about 50° which is considerably smaller compared to the original manuscript. Was there any change in the experiment protocol? Please clarify.

Responses 1: We appreciate your careful observation regarding the wrist extension angles stimulated by FES.

We attempted to place the FES electrodes in the same target muscle area as the previous experiment, but there may have been slight differences in positioning of the electrodes. That is why the wrist angle can be different from previous results. Additionally, the final joint angle induced by FES can vary based on the condition of the subjects on the day of the experiment. In spite of these differences, the consistent performance of the commercial and US-FES electrode across trials suggests that the experiment was implemented successfully.

Thank you for allowing us to clarify this aspect of our experimental protocol.

  1. In Figure 7, why did the authors use the joint angles calculated from angular velocity data? Since the authors have IMU data, the orientation of the IMU sensor can be calculated. Furthermore, it would be good to compare the orientations between the commercial FES electrodes and the developed US-FES electrodes. It is suggested to demonstrate the representative time-series angle data for each condition (i.e., the commercial vs. the developed electrodes) and to add more detailed characteristics of the time-series angle data (e.g., the delay between the stimulation and onset, the rate of angle change, etc.) as outcome measures in Table.

Thank you for your insightful question. The IMU we used in our experiments provides direct measurements of acceleration and angular velocity, but not joint angles. To obtain the joint angles, we integrated the angular velocity data over time. This approach allowed us to estimate the joint angles necessary for our analysis. We believe this method is robust and reliable for our purposes. We have included Table 2 next to Fig. 7 to show the summary of more detailed characteristics of the time-series angle data as you suggested.

Furthermore, we added other time-series angle data such as the time delay between the onset of FES and the start of muscle extension as well as the rate of angle change to Figure 8. The p value plots are included in figure 8. (b) and (c), respectively.  In Figure 8, we used asterisk (*) to show the p value. * means that p value represents less than 0.05, ** less than 0.01, and *** less than 0.001.

Thank you so much for making the manuscript plausible.

Responses 3: We appreciate the reviewer’s suggestion to reorganize Figure 8. We revised the figure as suggested. We added an Asterisk to Figure 8 (a) (Now, Figure 8) and removed Figure 8 (b).
